# Systemic Uremic Toxin Burden in Autism Spectrum Disorder: A Stratified Urinary Metabolite Analysis

**DOI:** 10.3390/ijms26157070

**Published:** 2025-07-23

**Authors:** Joško Osredkar, Teja Fabjan, Uroš Godnov, Maja Jekovec-Vrhovšek, Joanna Giebułtowicz, Barbara Bobrowska-Korczak, Gorazd Avguštin, Kristina Kumer

**Affiliations:** 1Institute of Clinical Chemistry and Biochemistry, University Medical Centre Ljubljana, Zaloška Cesta 2, 1000 Ljubljana, Slovenia; josko.osredkar@kclj.si (J.O.); teja.fabjan@kclj.si (T.F.); 2Faculty of Pharmacy, University of Ljubljana, Aškerčeva 7, 1000 Ljubljana, Slovenia; 3The Faculty of Mathematics, Natural Sciences and Information Technologies, University of Ljubljana, Glagoljaška ulica 8, 6000 Koper, Slovenia; uros.godnov@gmail.si; 4Center for Autism, Unit of Child Psychiatry, University Children’s Hospital, University Medical Centre Ljubljana, 1000 Ljubljana, Slovenia; maja.jekovec@kclj.si; 5Department of Bioanalysis and Drug Analysis, Faculty of Pharmacy with the Laboratory Medicine Division, Medical University of Warsaw, Banacha 1, 02-097 Warsaw, Poland; joanna.giebultowicz@wum.edu.pl; 6Department of Toxicology and Food Science, Faculty of Pharmacy with the Laboratory Medicine Division, Medical University of Warsaw, Banacha 1, 02-097 Warsaw, Poland; barbara.bobrowska@wum.edu.pl; 7Department of Microbiology, Biotechnical Faculty, University of Ljubljana, Groblje 3, 1230 Domžale, Slovenia; gorazd.avgustin@bf.uni-lj.si

**Keywords:** autism spectrum disorder, uremic toxins, p-cresyl sulfate, indoxyl sulfate, metabolomics, urinary biomarkers, gut microbiota, TMAO, ADMA

## Abstract

Autism spectrum disorder (ASD) is increasingly associated with microbial and metabolic disturbances, including the altered production of gut-derived uremic toxins. We investigated urinary concentrations of five representative uremic toxins—indoxyl sulfate (IS), p-cresyl sulfate (PCS), trimethylamine N-oxide (TMAO), asymmetric dimethylarginine (ADMA), and symmetric dimethylarginine (SDMA)—in 161 children with ASD and 71 healthy controls. Toxins were measured using LC-MS/MS and were normalized to creatinine. Subgroup analyses were performed by sex, age group (2–5.9 vs. 6–17 years), and autism severity based on the Childhood Autism Rating Scale (CARS). In addition to individual concentrations, we calculated the total toxin burden, proportional contributions, and functional ratios (IS/PCS, PCS/TMAO, and IS/ADMA). While individual toxin levels did not differ significantly between groups, stratified analyses revealed that PCS was higher in girls and in severe cases of ASD, whereas IS and TMAO were reduced in younger and more severely affected children. The functional ratios shifted consistently with severity—IS/PCS declined from 1.69 in controls to 0.99 in severe cases of ASD, while PCS/TMAO increased from 12.2 to 20.5. These patterns suggest a phenolic-dominant microbial signature and an altered host–microbial metabolic balance in ASD. Functional toxin profiling may offer a more sensitive approach to characterizing metabolic disturbances in ASD than concentration analysis alone.

## 1. Introduction

Autism spectrum disorder (ASD) is a heterogeneous neurodevelopmental condition characterized by impaired social communication, restricted interests, and repetitive behaviors [1,2].

Affecting approximately 1% of children globally, ASD is more frequently diagnosed in males than females, with a male-to-female ratio of around 4:1 [3,4].

Defective sensory processing, persistent/stereotypical behaviors, limited interests, and challenges with social interaction and communication are the primary indicators of ASD. Furthermore, ASD frequently co-occurs with intellectual disabilities (IDs), anxiety disorders, and hyperactivity [5].

ASD is a complex neurodevelopmental condition with a multifaceted etiology involving both genetic and environmental factors [6]. Although the etiology of ASD is not yet fully understood, both genetic and environmental factors have been implicated. Emerging evidence supports the role of the gut–brain axis in ASD, particularly in individuals with gastrointestinal (GI) symptoms, who often exhibit more severe behavioral impairments. Studies have demonstrated that individuals with ASD frequently exhibit altered intestinal permeability, gut dysbiosis, and microbial metabolite profiles [7,8].

An increasing number of studies have suggested that the gut microbiota–brain axis may have an effect on ASD. First, ASD has been associated with increased intestinal permeability and gastrointestinal (GI) problems. Second, children with ASD and concomitant GI symptoms exhibit more severe social impairments, anxiety, and sensory over-reactivity than their counterparts with ASD who do not have GI symptoms. Third, those with ASD present with a dysbiosis of the gut microbiota [9,10,11,12].

It is commonly known that gut-derived chemicals can enter the bloodstream from the gut, impact multiple systems, including the central nervous system, and then be filtered into urine in varying quantities at the glomerular level [13].

Uremic toxins are small molecules produced by host and microbial metabolism, particularly from the degradation of dietary amino acids. Among these, indoxyl sulfate (IS), p-cresyl sulfate (PCS), and trimethylamine N-oxide (TMAO) are primarily derived from the gut microbiota, while asymmetric and symmetric dimethylarginine (ADMA and SDMA) originate endogenously. These toxins can exert systemic effects through mechanisms involving oxidative stress, inflammation, and endothelial dysfunction. The production of uremic toxins from dietary substrates, especially amino acids like tryptophan, tyrosine, and choline, is largely dependent on gut bacteria. In particular, IS, PCS, and TMAO have the ability to build up in the bloodstream and affect host systems including the kidneys and brain. A schematic overview of the systemic and microbiological pathways associated with these metabolites is shown in Figure 1, emphasizing the possibility of imbalance in ASD. Using this methodology, we sought to evaluate children with ASD’s composite burden and relative functional correlations in addition to their specific toxin levels.

According to recent studies, there is a strong link between uremic toxins and ASD. Compared to healthy controls, children with ASD had higher urine levels of uremic toxins, especially PCS and IS [14,15]. Both core and comorbid symptoms of ASD may be enhanced by these toxins, which are produced by bacterial metabolism [15]. Metabolomic investigations have shown that children with ASD have an altered gut microbiota, elevated oxidative stress, and aberrant amino acid metabolism [16,17].

ADMA and SDMA were identified by Tain and Hsu (2017) as harmful amino acid derivatives that prevent the synthesis of nitric oxide and predict cardiovascular events in a number of illnesses [18]. In addition, Kielstein et al. described SDMA as a marker of renal function and a possible cause of inflammation and atherosclerosis, while ADMA is a strong nitric oxide synthase inhibitor and cardiovascular risk predictor [19].

In Table 1, the key information about ADMA, SDMA, TMAO, IS, and PCS as uremic toxins are summarized, as well as their potential implications in ASD.

Previous research has reported elevated urinary levels of IS and PCS in individuals with ASD, but the findings remain inconsistent due to methodological differences and population heterogeneity. Most studies have focused on absolute toxin concentrations, potentially overlooking more subtle shifts in metabolic balance [14,20,21,22,23]. Results frequently conflict each other because of differences in age, sex, and symptom severity. Furthermore, analyzing these toxins separately could mask underlying changes in metabolism.

In our study, we aimed to characterize the urinary profiles of five representative uremic toxins (IS, PCS, TMAO, ADMA, and SDMA) in children with ASD and healthy controls. Beyond individual concentrations, we explored total toxin burden, proportional contributions, and functional ratios (e.g., IS/PCS and PCS/TMAO) to identify the metabolic patterns associated with age, sex, and ASD severity. We hypothesized that ratio-based and composite assessments would reveal microbiota-related dysregulation that is not captured by concentration data alone.

The present study was undertaken to investigate potential factors contributing to elevated urinary uremic toxins levels in ASD. We thus measured total urinary uremic toxins in a newly recruited sample of 161 Slovene ASD children compared to a control group of 71 healthy children. We compared these results by age, sex, and ASD severity. In the next step, we formed age- and sex-matched groups, also presenting the same data in these smaller groups. The secondary goal of this investigation was to examine whether the composite assessment of uremic toxins is more indicative of consistent metabolic changes in children with ASD. This was assessed using combined total burden and functional ratios. Additionally, we wanted to see if these trends varied by age, sex, and the degree of ASD symptoms as determined by the Childhood Autism Rating Scale (CARS).

## 2. Results

### 2.1. Uremic Toxin Concentrations in the Whole Cohort

A total of 161 children with ASD, as well as 71 healthy controls, were included. The median values of urinary uremic toxins (normalized to creatinine) are summarized in Table 2. Overall, PCS showed a trend toward higher values in ASD (median 44.11 vs. 37.74 µmol/mmol creatinine), although this was not statistically significant. IS levels were slightly lower in ASD than controls (median 57.60 vs. 63.80 µmol/mmol), while TMAO also showed lower concentrations in ASD.

Among the endogenous methylated arginines, ADMA showed a non-significant elevation in ASD (14.72 vs. 12.70 µmol/mmol; *p* = 0.19), while SDMA remained comparable between groups (30.16 vs. 32.06 µmol/mmol).

### 2.2. Age-Stratified Findings

In the younger subgroup (2–5.9 years) (Appendix A), TMAO levels were markedly lower in ASD children compared to controls (median 2.48 vs. 4.36 µmol/mmol; *p* = 0.06), pointing toward potential altered choline metabolism or dietary differences in younger ASD children. In contrast, the differences were milder in the 6–17 age group (Appendix A).

### 2.3. Sex-Based Differences

The sex-stratified analysis revealed notable differences in PCS levels, which were higher in girls with ASD than in boys (median 46.65 vs. 40.75 µmol/mmol) (Appendix A). IS levels also tended to be higher in girls. These patterns suggest sex-specific differences in the microbiota composition or toxin clearance mechanisms.

### 2.4. ASD Severity and Toxin Levels

Stratification by Childhood Autism Rating Scale (CARS) scores revealed a trend toward higher PCS in children with more severe ASD (CARS > 36.5), with medians rising from 49.66 (mild/moderate) to 53.37 µmol/mmol (severe). IS levels decreased slightly with increasing severity, suggesting a shift in microbial metabolism patterns (Appendix A). The proportion of children according to CARS score, sex, and age is shown in Table 3.

### 2.5. Total Uremic Toxin Burden, Proportions, and Functional Ratios

To better reflect the overall systemic load and metabolic patterns of uremic toxins in ASD, we calculated the total median burden (sum of five urinary toxin medians), the percentage contribution of each metabolite to this total, and several functional ratios across groups.

Compared to controls, children with ASD had a slightly higher total uremic toxin load (151.95 vs. 148.65 µmol/mmol creatinine); the highest values were found in children with milder CARS scores (<36; 158.07 µmol/mmol) and younger ASD children (2–5.9 years; 155.11 µmol/mmol). These tendencies imply a higher systemic exposure in situations of early or less-severe ASD, despite little absolute differences (Table 4).

The overall burden’s content differed greatly between groups. The largest contributor in healthy controls was IS (42.9%), with PCS (25.4%) coming in second. On the other hand, PCS became co-dominant, contributing 35.1% of the burden, whereas IS decreased to 34.6% in children with more severe ASD (CARS > 36.5). Consistent levels of ADMA and SDMA (Table 5) and a steady or decreased contribution of TMAO (1.7–2.1%) accompanied this change. According to these results, the severity of ASD is associated with a progressive microbial metabolic shift toward phenolic metabolism.

The compositional shift in uremic toxin burden is further visualized in Figure 2, which illustrates the percentage contribution of each metabolite to the total urinary burden across subgroups. As ASD severity increased, the relative dominance of IS diminished, while PCS became increasingly prominent. While the absolute concentration changes were modest, the compositional burden profile provided a clearer differentiation between groups. These findings support the utility of proportional and ratio-based analyses for revealing microbial–metabolic dysregulation in ASD.

To further describe metabolic interactions, functional ratios were extracted from important toxin pairings. A relative change from indolic to phenolic toxin dominance was shown by the IS/PCS ratio, which decreased from 1.69 in controls to 1.22 in mild/moderate ASD and 0.99 in severe ASD. In a similar vein, the PCS/TMAO ratio increased from 12.2 in controls to over 20 in children who were more severely affected, indicating either greater proteolytic fermentation or decreased methylamine metabolism. Additionally, the IS/ADMA ratio decreased from 5.02 to 3.55 as the severity of ASD increased, indicating a potential interaction between systemic vascular toxins and gut-derived toxins (Table 6).

The results shown in Table 6 are further illustrated in Figure 3, where derived functional ratios are plotted across control and ASD severity subgroups. A consistent decline in the IS/PCS ratio with increasing ASD severity suggests a progressive shift from indolic to phenolic microbial metabolism. Similarly, PCS/TMAO ratios were elevated in both mild and severe ASD, potentially reflecting increased proteolytic fermentation or decreased choline-derived methylamine conversion. The IS/ADMA ratio also declined across ASD subgroups, suggesting systemic metabolic interactions involving endothelial function and nitric oxide dysregulation. These findings support the use of composite ratios as sensitive biomarkers of metabolic shifts in ASD.

In line with the previous results demonstrating higher PCS levels in females, a sex-specific analysis (Table 7) showed a slightly higher total burden and PCS fraction in girls with ASD. Although girls displayed greater variability, the IS/PCS and PCS/TMAO ratios were similar for boys and girls. The constancy of the microbial–metabolic shift over development was further supported by age-stratified analysis (Table 8), which revealed that both younger and older ASD subgroups exhibited lower IS/PCS ratios and higher PCS/TMAO ratios when compared to age-matched controls. When combined, these compositional and ratio-based results imply that the functional assessment of uremic toxins offers a more complex picture of metabolic imbalance associated with ASD than concentration comparisons alone.

## 3. Discussion

Our findings provide new insights into the metabolic phenotype of children with ASD, highlighting modest but consistent differences in uremic toxin profiles compared to healthy controls. While individual toxin concentrations were not significantly altered, composite metrics—including total burden, proportional contributions, and functional ratios—revealed metabolic shifts associated with ASD severity, sex, and age.

### 3.1. Toxins Derived from the Microbiota and Dysbiosis Associated with ASD

The bacterial breakdown of dietary tryptophan and tyrosine produces the microbiota-derived uremic toxins IS and PCS, respectively [24]. The intestinal epithelium absorbs these substances, the liver metabolizes them, and renal tubular secretion removes them [25]. *Clostridium*, *Bacteroides*, and other proteolytic bacteria have been found to be more abundant in the gut microbiota of people with ASD, in line with earlier research [26,27]. Given that these microorganisms are known to promote the synthesis of phenolic and indolic metabolites, there may be a mechanism behind the observed rise in PCS burden.

The idea of microbial metabolic changes in ASD is supported by the increasing rise in PCS contribution from 25.4% in controls to 35.1% in children with more severe ASD (CARS > 36.5), as well as a declining IS/PCS ratio. These changes might favor routes for tyrosine metabolism (which produces PCS) over those for tryptophan metabolism (which yields IS). This could be because of changes in intestinal transit time, microbial enzyme expression, or substrate availability [28].

### 3.2. Systemic Inflammation and Methylarginines

Endogenous methylated arginine derivatives, such as ADMA and SDMA, have been associated with immunological dysregulation, oxidative stress, and endothelial dysfunction [29]. They also block nitric oxide synthase. The stable presence of ADMA and SDMA, even without statistical significance, suggests a background of low-grade systemic involvement, possibly linked to endothelial dysfunction and nitric oxide dysregulation in ASD. Even within normal bounds, a lower SDMA/ADMA ratio in children with ASD, particularly males, may indicate a disturbance in arginine metabolism or slight variations in renal function.

### 3.3. Total Burden and Shifting Proportions: Toward a Toxin Signature

Children with ASD had a somewhat higher overall burden than controls when the sum of their median toxin concentrations was considered (151.95 vs. 148.65 µmol/mmol creatinine). However, the changes in proportional contribution were more noticeable. IS was the largest contributor in controls (42.9%), but in ASD subgroups, especially those with higher CARS scores, its relative proportion dropped. PCS became the co-dominant toxin at the same time. Although there are not many significant variations in absolute concentration, these compositional changes point to a qualitative change in toxin profiles that might more accurately represent the metabolism unique to ASD than absolute numbers alone.

Interestingly, the PCS/TMAO ratio rose significantly with severity, from 12.2 in controls to over 20 in the children who were severely affected. Another microbiota-derived toxin, TMAO, which is generated from dietary carnitine and choline, either stayed mostly unchanged or slightly declined. These trends could be the result of changes in liver FMO activity, which changes trimethylamine (TMA) to TMAO, causes microbial shifts away from TMA producers, or changes dietary behaviors in ASD (such as decreased choline consumption or food selectivity) [30].

### 3.4. Effects Dependent on Age and Sex

The sex-stratified analysis indicated higher PCS and IS levels in girls with ASD, aligning with previous research suggesting sex-specific metabolic and microbial profiles. Similarly, reduced TMAO levels in younger children may reflect developmental differences in diet, microbiota, or hepatic enzyme activity [31]. Estrogens are known to interact with microbial profiles and may alter the activity of the sulfotransferase enzyme, which could affect the clearance of toxins.

Age-related variations were particularly noteworthy, whereby younger children with ASD (2–5.9 years) had considerably lower levels of TMAO than controls, a tendency that was not seen in older children. Age-related differences in food composition, hepatic FMO activity, and gut microbiota maturation may be the cause of this [32].

### 3.5. Clinical Consequences

The patterns of the uremic toxins, particularly the ratios and proportions, may be useful in classifying ASD subtypes or tracking the effectiveness of treatment, even though none of them alone are reliable biomarkers for ASD. We observed a progressive decline in the IS/PCS ratio and a rise in the PCS/TMAO ratio with increasing severity of ASD, pointing toward a shift from indolic to phenolic microbial metabolism. This may reflect underlying alterations in gut microbiota composition, proteolytic fermentation, or hepatic detoxification capacity. The elevated relative contribution of PCS in severe ASD supports the hypothesis of a phenolic-dominant microbial signature in this subgroup.

Furthermore, the presence of uremic toxins such as IS and ADMA, which are both associated with oxidative stress and endothelial dysfunction, suggests that some ASD phenotypes may be influenced by neurovascular mechanisms [33].

## 4. Materials and Methods

### 4.1. Participants

Urine samples were collected from children diagnosed with ASD (*n* = 161; 124 boys, 37 girls) and healthy controls (*n* = 71; 37 boys, 34 girls). The group consisted of children aged from 2 years to 17 years. In the first step, we formed a group of children aged between 2 and 5.9 years, in which we had 35 children with ASD and 14 healthy children (group code = A); the remainder was a group of children aged between 6 and 17 years, in which we had 126 children with ASD and 57 healthy children (group code = B). The data on the proportions by sex and age are given in Table 9.

### 4.2. Methods

Uremic toxins measured included IS, PCS, TMAO, ADMA, and SDMA using validated LC-MS/MS techniques. Concentrations were normalized to urinary creatinine. The CARS score was used to categorize ASD severity. Stool form was assessed using the Bristol Stool Chart (BSC). Data were statistically analyzed using appropriate non-parametric tests, with significance set at *p* < 0.05.

Quantification of TMAO, ADMA, and SDMA: MRM transitions along with the corresponding de-clustering potential (DP) and collision energy (CE) values were as follows: for ADMA, the transition *m*/*z* 203 → 46 was used (DP: 61 V, CE: 41 V); for SDMA, *m*/*z* 203 → 172 (DP: 61 V, CE: 19 V); for ADMA-d6, *m*/*z* 209 → 77 (DP: 66 V, CE: 45 V); for TMAO, *m*/*z* 76 → 42 (DP: 66 V, CE: 53 V); and for TMAO-d9, *m*/*z* 85 → 46 (DP: 61 V, CE: 59 V). Chromatographic separation was carried out using a SeQuant^®^ ZIC^®^-HILIC column (50 × 2.1 mm, 5 μm particle size; Merck, Darmstadt, Germany), maintained at 25 °C, with a mobile phase flow rate of 0.5 mL/min. The mobile phases consisted of 20 mM ammonium acetate (eluent A) and acetonitrile containing 0.2% formic acid (eluent B). The gradient elution profile (%B) was as follows: 0 min—95%, 1 min—95%, 7 min—50%, and 8 min—50%. A 5 μL volume of sample was injected. Urine samples (0.1 mL) were prepared by mixing with internal standards (0.1 mL of 6 μg/mL ADMA-d6 and TMAO-d9) and acetonitrile (0.6 mL), followed by vortexing for 3 min and centrifugation at 10,000× *g* for 5 min before LC analysis.

Quantification of PCS and IS: MRM transitions were as follows: for PCS, *m*/*z* 186.9 → 106.9, and for PCS-d7, *m*/*z* 194.0 → 114.0; for IS, *m*/*z* 211.9 → 79.8, and for IS-d4, *m*/*z* 216.0 → 79.9. Instrument settings including DP, CE, EP, and CXP were as follows—for IS and IS-d4: −60, −38, −10, −5 V and −65, −46, −10, −1 V, respectively; for PCS and PCS-d7: −65, −28, −10, −7 V and −60, −30, −10, −7 V, respectively. Separation was performed using a Kinetex C18 column (100 × 4.6 mm, 2.6 μm; Phenomenex, Torrance, CA, USA), held at 40 °C with a flow rate of 0.5 mL/min. The mobile phases included water with 0.1% formic acid (A) and methanol with 0.1% formic acid (B). The gradient program was as follows: 0 min—10% B, 0.5 min—10% B, 4.5 min—95% B, 8.5 min—95% B. The injection volume was 10 μL. Prior to LC injection, urine (0.01 mL) was mixed with 0.05 mL of internal standards (2 μg/mL PCS-d7 and IS-d4) and 0.6 mL of methanol, vortexed for 3 min, and centrifuged for 5 min at 10,000 g, before being subsequently diluted sixfold with water.

### 4.3. Statistics

R version 4.3.1 in conjunction with RStudio version 2023.12.0 was used for statistical analysis, using the tidy verse suite [34] for visualization and arsenal package [35] to compare groups.

The Shapiro–Wilk tests from the stats package [36,37] was used to evaluate the distribution of data. The results showed that all datasets had non-normal distributions; therefore, the non-parametric Wilcoxon signed-rank test was employed for comparisons. The Benjamini–Hochberg procedure was used to control false discovery rate, with alpha significance level of 0.05.

## 5. Conclusions

In our study, we analyzed urinary uremic toxins in a large cohort of children with ASD compared to healthy controls. Although absolute toxin levels showed limited group differences, we identified consistent trends when considering functional ratios and proportional contributions. Specifically, we observed a decrease in the IS/PCS ratio and an increase in the PCS/TMAO ratio with increasing ASD severity, suggesting a phenolic-dominant microbial signature.

Higher PCS and IS levels were also noted in girls with ASD, while younger children showed reduced TMAO concentrations. These findings suggest sex- and age-specific differences in microbiota-related metabolism.

Despite the lack of significant group differences, the consistent presence of methylated arginines (ADMA and SDMA) and the trend toward higher levels of ADMA in girls with ASD may suggest low-grade systemic or vascular involvement, which is consistent with the idea that nitric oxide dysregulation plays a role in the pathophysiology of ASD.

Our results support the use of toxin ratios and compositional profiles as sensitive indicators of microbial–host metabolic interactions in ASD.

Limitations: Our study has several limitations. First, its cross-sectional design prevents causal interpretation. Second, dietary intake and medication use, which may influence metabolite levels, were not controlled. Third, the number of girls and younger children in control groups was limited, reducing statistical power for subgroup analyses. Finally, we did not assess microbiota composition directly, which limits mechanistic conclusions.

Inflammatory markers and PINI: Given the known link between systemic inflammation and uremic toxin generation, future studies should integrate inflammatory and nutritional biomarkers such as the Prognostic Inflammatory and Nutritional Index (PINI). This index has shown promise in the early detection of inflammation in other clinical settings [38] and may offer complementary insights in relation to ASD research.

Taken together, our results support the utility of composite toxin metrics in characterizing microbiota-related metabolic changes in ASD and suggest potential value in including inflammatory markers in future stratification models.

## Figures and Tables

**Figure 1 ijms-26-07070-f001:**
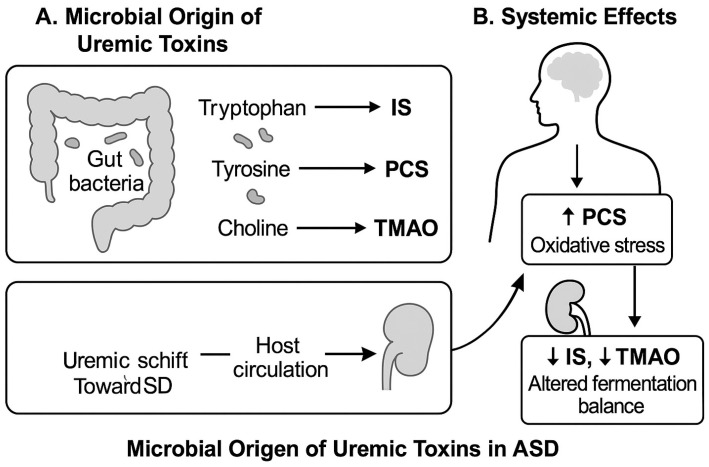
Microbial origins and systemic impacts of uremic toxins in autism spectrum disorder. Schematic overview illustrating the gut microbiota-mediated production of key uremic toxins—indoxyl sulfate (IS), p-cresyl sulfate (PCS), and trimethylamine N-oxide (TMAO)—from dietary amino acids (tryptophan, tyrosine, and choline). These toxins enter systemic circulation and may contribute to oxidative stress, inflammation, and neurological disturbances through interactions with the kidneys and brain. This functional pathway supports the rationale for studying toxin ratios and burden in ASD.

**Figure 2 ijms-26-07070-f002:**
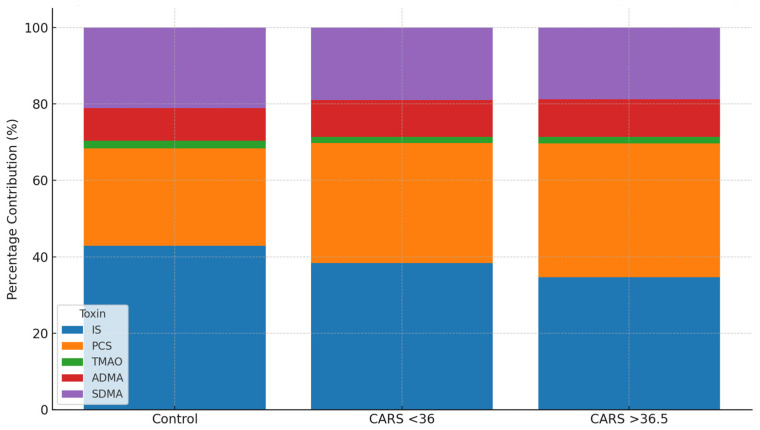
Relative contribution of uremic toxins to total burden by ASD severity. Stacked bar graph showing the percentage contribution of five urinary uremic toxins—IS, PCS, TMAO, ADMA, and SDMA—to the total burden in control and ASD subgroups stratified by severity (CARS < 36 and CARS > 36.5). With increasing ASD severity, the relative contribution of PCS increased, while IS declined, indicating a compositional shift toward phenolic toxin dominance. TMAO remained low across groups, and proportions of methylarginines were relatively stable.

**Figure 3 ijms-26-07070-f003:**
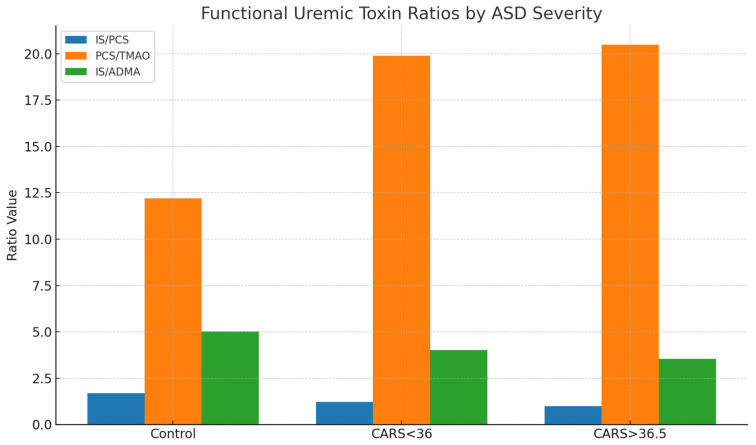
Functional uremic toxin ratios according to ASD severity. Bar chart showing three derived urinary toxin ratios—IS/PCS, PCS/TMAO, and IS/ADMA—in healthy controls, children with mild/moderate ASD (CARS < 36), and those with more severe ASD (CARS > 36.5). With increasing ASD severity, the IS/PCS and IS/ADMA ratios progressively declined, while the PCS/TMAO ratio increased, indicating a phenolic-dominant microbial shift and reduced methylamine metabolism. These ratio-based changes provide deeper insight into functional metabolic imbalance than absolute toxin concentrations alone (see also Table 6).

**Table 1 ijms-26-07070-t001:** Description of uremic toxins with potential implications in ASD.

Uremic Toxin	Description	Potential Implications in ASD
ADMA (Asymmetric Dimethylarginine)	Endogenous inhibitor of nitric oxide synthase (NOS); associated with endothelial dysfunction and cardiovascular complications in CKD	Elevated levels may contribute to endothelial dysfunction and impaired nitric oxide signaling; potential impact on neurodevelopmental processes in ASD
SDMA (Symmetric Dimethylarginine)	Related to ADMA; accumulates in CKD and associated with endothelial dysfunction and cardiovascular disease	Limited specific research linking SDMA to ASD; potential role in endothelial dysfunction and cerebral blood flow
TMAO (Trimethylamine N-Oxide)	Derived from gut microbial metabolism of dietary choline, phosphatidylcholine, and carnitine; associated with cardiovascular disease and altered gut microbiota	Limited direct evidence linking TMAO to ASD; dysregulation of gut microbiota and neuroinflammation could be relevant
IS (Indoxyl Sulfate)	Metabolite of tryptophan produced by gut bacteria; accumulates in CKD and excreted in urine	Elevated levels of IS may impact behavior and cognition; potential association with ASD symptoms
PCS (P-Cresol Sulfate)	Metabolite of tyrosine and phenylalanine produced by gut bacteria; accumulates in CKD and excreted in urine	Some evidence suggests a potential link between elevated PCS levels and ASD-like behaviors; further research needed to establish causality

**Table 2 ijms-26-07070-t002:** Concentrations of uremic toxins normalized to creatinine in the whole group of ASD and healthy children.

	Control (N = 71)	ASD (N = 161)	ASD/Control Ratio	*p* Value
ADMA				0.19
Mean (SD)	14.23 (7.03)	15.56 (6.93)	1.09	
Median (Q1, Q3)	12.70 (9.73, 18.74)	14.72 (11.19, 19.24)	1.15	
Min–Max	1.67–33.16	2.50–34.36		
SDMA				0.93
Mean (SD)	33.00 (14.98)	34.01 (16.11)	1.03	
Median (Q1, Q3)	32.06 (23.46, 39.28)	30.16 (23.66, 44.29)	0.94	
Min–Max	7.07–80.21	4.09–79.83		
TMAO				0.30
Mean (SD)	3.52 (2.72)	3.05 (2.48)	0.87	
Median (Q1, Q3)	3.09 (1.48, 5.19)	2.65 (1.06, 4.37)	0.86	
Min–Max	0.00–10.38	0.01–10.52		
IS				0.23
Mean (SD)	71.04 (40.71)	64.08 (39.90)	0.90	
Median (Q1, Q3)	63.80 (39.77, 103.18)	57.60 (30.19, 87.37)	0.90	
Min–Max	3.57–168.60	3.16–193.60		
PCS				0.86
Mean (SD)	52.82 (43.65)	56.68 (47.56)	1.07	
Median (Q1, Q3)	37.74 (20.02, 80.02)	44.11 (15.73, 84.23)	1.17	
Min–Max	1.27–195.67	0.13–188.20		

The results are in µmol/mmol of creatinine.

**Table 3 ijms-26-07070-t003:** Data on the proportions by sex and age for different CARS scores.

	Control (N = 71)	CARS < 36 (N = 58)	CARS > 36.5 (N = 34)
SEX			
Boys	37 (52.1%)	45 (77.6%)	27 (79.4%)
Girls	34 (47.9%)	13 (22.4%)	7 (20.6%)
AGE (years)			
Mean (SD)	8.93 (3.82)	9.38 (3.92)	9.72 (3.87)
Median (Q1, Q3)	8.60 (6.20, 11.25)	8.70 (6.12, 11.82)	9.00 (6.40, 13.00)
Min–Max	2.40–16.70	2.50–17.00	3.50–16.70

**Table 4 ijms-26-07070-t004:** Total uremic toxin burden (sum of median values).

Group	Total Burden
Controls	148.65
ASD Overall	151.95
ASD 2–5.9 y	155.11
ASD 6–17 y	153.35
CARS < 36	158.07
CARS > 36.5	151.91

Total burden calculated as the sum of median concentrations (µmol/mmol creatinine) of IS, PCS, TMAO, ADMA, and SDMA for control and ASD subgroups.

**Table 5 ijms-26-07070-t005:** Percentage contribution of each toxin to total burden.

Group	IS%	PCS%	TMAO%	ADMA%	SDMA%
Control	42.9	25.4	2.1	8.5	21.1
CARS < 36	38.4	31.4	1.6	9.6	19.0
CARS > 36.5	34.6	35.1	1.7	9.8	18.8

Relative contribution (%) of each toxin (based on median values) to the total uremic burden in control and ASD subgroups.

**Table 6 ijms-26-07070-t006:** Key ratios between uremic toxins.

Group	IS/PCS	PCS/TMAO	IS/ADMA
Control	1.69	12.2	5.02
CARS < 36	1.22	19.9	4.01
CARS > 36.5	0.99	20.5	3.55

Calculated functional ratios based on medians of selected toxins in control and ASD subgroups.

**Table 7 ijms-26-07070-t007:** Sex-specific total burden and ratios (ASD only).

Sex	Total Burden	IS/PCS	PCS/TMAO
Boys	153.52	1.32	17.7
Girls	156.13	1.24	18.1

Sex-specific comparison of total toxin burden, as well as IS/PCS and PCS/TMAO ratios, in ASD children.

**Table 8 ijms-26-07070-t008:** Ratios by age group (ASD vs. controls).

Group	IS/PCS	PCS/TMAO
Control (2–5.9 years)	1.80	13.3
ASD (2–5.9 years)	1.36	20.0
Control (6–17 years)	1.68	11.7
ASD (6–17 years)	1.37	19.5

Comparison of key toxin ratios across younger (2–5.9 years) and older (6–17 years) children with ASD and matched controls.

**Table 9 ijms-26-07070-t009:** Data on the proportions by sex and age.

**2–17 Years**	**Control (N = 71)**	**ASD (N = 161)**	**Total (N = 232)**
SEX			
Boys	37 (52.1%)	124 (77.0%)	161 (69.4%)
Girls	34 (47.9%)	37 (23.0%)	71 (30.6%)
AGE			
Mean (SD)	8.93 (3.82)	9.28 (3.76)	9.17 (3.77)
Median (Q1, Q3)	8.60 (6.20, 11.25)	8.60 (6.20, 12.40)	8.60 (6.20, 12.10)
Min–Max	2.40–16.70	2.50–17.00	2.40–17.00
**2–5.9 years**	**Control A (N = 14)**	**ASD A (N = 35)**	**Total A (N = 49)**
SEX			
Boys	10 (71.4%)	23 (65.7%)	33 (67.3%)
Girls	4 (28.6%)	12 (34.3%)	16 (32.7%)
AGE			
Mean (SD)	3.85 (1.15)	4.57 (0.92)	4.37 (1.03)
Median (Q1, Q3)	4.10 (2.68, 4.65)	4.60 (3.95, 5.40)	4.40 (3.80, 5.40)
Min–Max	2.40–5.50	2.50–5.90	2.40–5.90
**6–17 years**	**Control B (N = 57)**	**ASD-B (N = 126)**	**Total B (N = 183)**
SEX			
Boys	27 (47.4%)	101 (80.2%)	128 (69.9%)
Girls	30 (52.6%)	25 (19.8%)	55 (30.1%)
AGE			
Mean (SD)	10.18 (3.15)	10.59 (3.15)	10.46 (3.15)
Median (Q1, Q3)	10.40 (7.20, 12.40)	10.35 (7.93, 13.00)	10.40 (7.70, 12.95)
Min–Max	6.00–16.70	6.00–17.00	6.00–17.00

## Data Availability

The data that support the findings of this study are available from the study’s principal investigator—O.J.—upon reasonable request.

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
