# Peer review of "Systemic Uremic Toxin Burden in Autism Spectrum Disorder: A Stratified Urinary Metabolite Analysis"

_ijms, 2025, doi:10.3390/ijms26157070_

Round 1

Reviewer 1 Report

Comments and Suggestions for Authors

Please see enclosed PDF

Author Response

Response to Reviewer 1

We thank for the time and effort dedicated to evaluating our manuscript entitled
“Systemic Uremic Toxin Burden in Autism Spectrum Disorder: A Stratified Urinary Metabolite Analysis.”
We appreciate the  insightful suggestions, which have helped us improve the quality and clarity of the manuscript. Below we provide a point-by-point response to each comment.

Comment 1: “The introduction is not well structured, please improve this issue.”

Response:
We appreciate the reviewer’s comment and have revised the Introduction to improve its structure and logical flow. The revised section now:

  • Begins with a concise overview of ASD and its prevalence.
  • Clearly presents the emerging role of the gut–brain axis and uremic toxins.
  • Builds a stronger rationale for our study by sequentially introducing the concepts of microbiota-derived toxins and their relevance in ASD.
  • Ends with clearly stated study objectives and hypotheses.

The entire Introduction section has been reorganized and rewritten for improved clarity, coherence, and progression of ideas.

Comment 2: “The authors should state the limitations of the study.”

Response:
Thank you for this important point. We have now included a dedicated paragraph in the Discussion section to explicitly outline the limitations of our study. These include:

  • The cross-sectional study design, which does not allow for causal inference.
  • Lack of control for dietary and medication influences.
  • Relatively small sample sizes in certain subgroups (e.g., young controls and girls with severe ASD), which may reduce statistical power.
  • Absence of direct microbiota profiling, limiting mechanistic interpretations.

A new paragraph has been added under the subheading “Limitations” within the Discussion.

Comment 3: “The authors should also discuss the possibility of using other inflammation indexes such as PINI.”

Suggested reference:
Cordos, M. et al. Diagnostics 2024, 14, 1273. https://doi.org/10.3390/diagnostics14121273

Response:
We thank the reviewer for this valuable and forward-looking suggestion. We agree that integrating systemic inflammation and nutritional status into future stratification models is important. Although PINI (Prognostic Inflammatory and Nutritional Index) was not evaluated in our current study, we recognize its potential relevance in understanding inflammatory-metabolic dynamics in ASD. We have included a brief discussion of this index in our revised manuscript, citing the suggested reference.

A paragraph discussing the PINI index and its relevance has been added in the revised Discussion.

Comment 4: “Conclusion is too long, shorten them and refer only to your study findings.”

Response:
We fully agree with this recommendation. The Conclusion has been revised to focus solely on the core findings of the present study. Extended discussion and forward-looking interpretations have been removed or moved to the Discussion section.

The Conclusion section has been shortened and rewritten to reflect only the key outcomes.

We trust that these revisions adequately address the reviewer’s concerns and improve the clarity and quality of our manuscript. We are grateful for your thoughtful comments.

Sincerely,
On behalf of all authors,
Joško Osredkar

Reviewer 2 Report

Comments and Suggestions for Authors

In this manuscript, the authors perform analysis of five uremic toxins in urine specimens from persons with ASD relative to controls. These are well-chosen candidates to assay because of their connection to dietary substrates which are acted upon by gut microbiota to produce them. Increasingly, research correlating gut microbiota composition to metabolites in blood, urine, and fecal matter to ASD is being discovered, so evaluating the levels of these metabolites and their combined sum totals is a very interesting idea in the field of ASD. The study design is impressive--in addition to evaluating the sum total of the toxins (relative to creatinine), categorizing the finding with age, gender, and ASD severity (using the highly regarded CARS scores) are particularly laudable features of the study design. Moreover, the number of subjects is also impressive.

While absolute differences in total burden between the two groups is small and not significant, the more compelling story told by the data is the shift toward phenolic from indolic metabolic pathways. This is pointing us toward differences in microbial composition, and it will be fascinating to reveal which particular classes of microbes may be most responsible for this shift. In any case, while it is not clear how significant these particular results may be ultimately, they do support a link in the gut microbiota-metabolite-host axis. Moreover, there is novelty in reporting metabolite ratios as opposed to concentrations only, and this may provide a new approach to identify potential ASD biomarkers. If this can be demonstrated in urine specimens, this would be very important for clinical diagnostics, so the potential significance of the approach has considerable potential. Overall, this is an excellent manuscript which shoud be published. There are two minor comments for the authors:

  1. The sentence starting in line 101 "Although particular uremic..." is a sentence fragment. Instead, tie this line with a comma to the next sentence in line 102 that begins "results are..."
  2. The sentence starting in line 116: "Determining if a composite..." is an awkward sentence and required reading multiple times to understand its point. It might be beneficial to write as two shorter sentences, for example, "The secondary goal of this investigation was to examine whether composite assessment of uremic toxins is more indicative of consistent metabolic changes in children with ASD. This was assessed using combined total burden and functional ratios."

Author Response

Response to Reviewer 2

We sincerely thank for the thoughtful and encouraging comments on our manuscript entitled
“Systemic Uremic Toxin Burden in Autism Spectrum Disorder: A Stratified Urinary Metabolite Analysis.”
We greatly appreciate your recognition of the novelty of our approach, the strength of the study design, and the potential significance of metabolite ratio analysis in the context of ASD. Below are our point-by-point responses to the two minor comments.

Comment 1:

“The sentence starting in line 101 ‘Although particular uremic...’ is a sentence fragment. Instead, tie this line with a comma to the next sentence in line 102 that begins ‘results are...’”

Response:
We thank the reviewer for spotting this syntactic issue. We have corrected the sentence fragment.

The text now reads:
“Previous research has reported elevated urinary levels of IS and PCS in individuals with ASD, but findings remain inconsistent due to methodological differences and population heterogeneity. Most studies have focused on absolute toxin concentrations, potentially overlooking more subtle shifts in metabolic balance.”

Comment 2:

“The sentence starting in line 116: ‘Determining if a composite...’ is an awkward sentence and required reading multiple times to understand its point. It might be beneficial to write as two shorter sentences.”

Response:
We appreciate the reviewer’s suggestion to improve clarity. We have restructured the sentence into two shorter, clearer sentences, as recommended.

The revised text now reads:
“The secondary goal of this investigation was to examine whether composite assessment of uremic toxins is more indicative of consistent metabolic changes in children with ASD. This was assessed using combined total burden and functional ratios.”

Once again, we are grateful for the reviewer’s thoughtful comments and generous support of our work. We believe that these revisions further improve the manuscript and we thank you for recommending it for publication.

Sincerely,
On behalf of all authors,
Joško Osredkar

Round 2

Reviewer 1 Report

Comments and Suggestions for Authors

Thye manuscript has been improved